# FEDMODN: FEDERATED MODULAR DECISION SUPPORT NETWORKS
## *Learning from Imperfectly Interoperable Distributed Data*

**Cécile Trottet**[*]
University of Zurich
Zurich, Switzerland

**Michael Krauthammer**
University of Zurich
Zurich, Switzerland

**Mary-Anne Hartley**
EPFL
Lausanne, Switzerland

## ABSTRACT

This work proposes *FedMoDN*, a novel federated modular neural network architecture for collaborative learning across all features of an imperfectly interoperable distributed dataset. Here, distributed data centers that collect variable combinations of features are able to use the full complement of their features with minimal exposure to biased missingness. Our approach enables data owners collecting different feature subsets to train a joint model without sharing, discarding, or imputing any data. We evaluate the robustness of our approach through experiments that mirror realistic challenges encountered with medical data, particularly in resource-limited settings. Our results show that this modular approach is significantly more robust than a monolithic neural network when dealing with missing data, systematic bias, or heterogeneous feature subsets.

## 1 BACKGROUND

Collaborative learning approaches can greatly benefit clinical environments where pooling the insights of several distributed small databases can improve the effectiveness of local models. However, learning across fragmented data becomes complex when there is a high amount of heterogeneity across data sets. Particularly challenging is the case of imperfect interoperability (IIO) where nodes collect different feature subsets, and are thus vertically partitioned. This scenario is particularly relevant in clinical environments where resource limitations or variations in clinical practices and protocols can result in a high degree of systematically missing features. Another major source of bias stems from the way clinical data is often collected as a byproduct of decision-support tools. This can introduce spurious correlations between feature availability and outcome values, particularly when decision-tree-based tools are used.

In this work, we propose to address these challenges by adapting Modular Decision Support Networks (*MoDN*) (Trottet et al., 2023), a modular neural network architecture to the federated learning (FL) setting. This allows clinicians working with IIO clinical variable subsets to collaboratively train a model without discarding incomplete patient records or relying on feature imputation. We show its ability to match and outperform traditional FL, especially in the face of systematic bias. Although designed for low-resource medical environments, our approach is use case agnostic and applicable across domains.

### 1.1 RELATED WORK

Modular deep learning (DL) architectures are promising for transfer and multi-task learning by enabling composability, re-usability, and personalization of model components. For a taxonomy and recent survey of modular deep learning architectures, see Pfeiffer et al. (2023). In federated learning (McMahan et al., 2017a), recent work has focused on personalization to tackle data heterogeneity between nodes. Arivazhagan et al. (2019) introduced "personalization layers", which are private model components tailored to local data within shared model architectures. Similarly, Wang et al. (2022) proposed to adaptively select personalized modules from a federated pool of modules. Their

---

[*]cecileclaire.trottet@uzh.ch

work focuses on identifying optimal module combinations to build heterogeneous neural architectures. In contrast, our approach is based on modular decision support networks (MoDN (Trottet et al., 2023; Swamy et al., 2024)), which allows users to compose modules based on the specific feature subsets they wish to use, with a focus on interpretability by design, making the architecture particularly well-suited for medical applications.

## 1.2 CONTRIBUTIONS

Our work explores the potential of modular networks for collaborative learning in settings with sparse, biased, and IIO distributions. In particular, to incentivize collaboration, we prioritize majority robustness, aiming to mitigate the adverse effects of isolated data poisoning and shifts in minority datasets. To this end, we adapt the *MoDN* architecture (Trottet et al., 2023), which is modular across both input and output spaces, and demonstrate how it can be trained in a federated way to address the challenge of vertical partitioning in IIO datasets. Our model, *FedMoDN*, is composable, allowing the contributing hospitals/nodes to train and use only the modules relevant to their local dataset, without requiring any imputation for missing features or modalities. This is a significant improvement over standard horizontal federated learning (HFL) approaches, where incomplete datapoints must be discarded or imputed. Even more challenging is the mixed horizontal and vertical federated learning (HVFL), i.e. scenarios where different hospitals may have some overlap in patient IDs but distinct features (Figure 1, Panel 2.). This typically requires that non-overlapping IDs and features be discarded. *FedMoDN* allows learning from the entire dataset. We create a proof-of-concept using well-established FL paradigms, such as centralized server aggregation. However, our approach is flexible and can be readily adapted to other FL frameworks.

## 2 METHODS

### 2.1 MODEL ARCHITECTURE

We provide a brief overview of the model architecture and refer to Trottet et al. (2023); Swamy et al. (2024) for more specific details. As shown in Figure 1 Panel 3. the architecture relies on three main components: a state vector, a set of encoder modules, and a set of decoder modules. The state vector represents the current internal model representation learned about a patient. For each available feature $f_1, \ldots, f_n$, we define a corresponding encoder multilayer perceptron (MLP). Note that different encoder types could be defined for different modalities, such as convolutional neural networks for images. Each encoder module takes the corresponding feature value and the current patient state as input, and outputs the updated state. Similarly, we define one MLP decoder module for each target. The decoders take the current state as input and output the target prediction.

### 2.2 TRAINING

We describe the training procedure for *FedMoDN*, which involves two main steps: local module parameter updates and federated module parameter updates. In the federated step, we consider two scenarios: (1) local datasets with entirely distinct (non-overlapping) patients/training instances, and (2) local datasets with partial or complete overlap. While we outline a potential approach for training in the overlapping scenario, our experiments currently focus on the non-overlapping case, leaving identifying optimal strategies for training with overlap for future work.

#### 2.2.1 LOCAL PARAMETER UPDATES

Each node defines sets of encoder and decoder modules based on their local availability of features and targets. Specific architecture details such as the number or size of layers of each module are predefined by the centralized server to ensure compatibility during federated training. After initializing the state vector to zero, the encoders are applied sequentially in random order. Each time a feature is encoded, the state is updated, and all decoders are applied to predict their respective targets. *This trains the model to learn to predict the targets for any given set of available features, without requiring missing value imputation.*

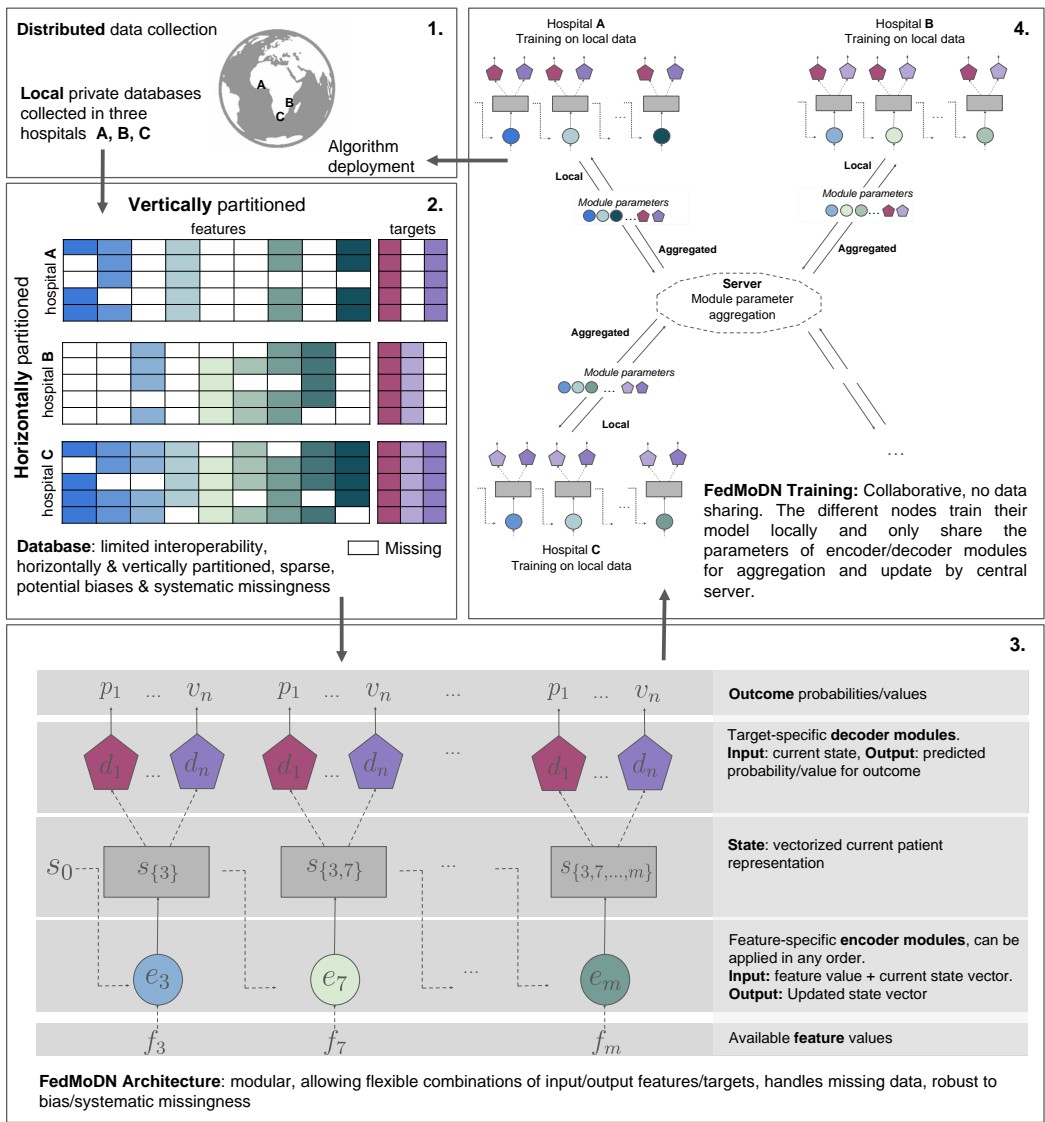

Figure 1: *FedMoDN* implementation pipeline. **Panel 1.** Decentralized data collection and algorithm deployment by various hospitals/nodes. **Panel 2.** The aggregated database is both horizontally partitioned (different patients) and vertically partitioned (different features/targets). Furthermore, it can be sparse and contain systematic biases. We assume each node collects at least one target per patient. **Panel 3.** *FedMoDN* architecture. Starting with the initial patient state vector $s_0$ (current internal patient representation) the state is sequentially updated by feature-specific encoder neural networks, which take the current state and the feature value as inputs. If a feature is missing, its encoder is not applied. After each state update, any target-specific decoder neural network can be used to generate predictions. **Panel 4.** Federated training of *FedMoDN* without training instance overlap. Each node trains only the modules corresponding to its available features, performing a predefined number of local training steps before sending module parameters to the central server. The central server aggregates the module parameters for shared features/targets and redistributes them for the next training round.

### 2.2.2 HORIZONTAL AND VERTICAL FL **WITHOUT** TRAINING INSTANCE OVERLAP

After completing a predefined number of local training steps and updating their encoder and decoder module parameters, the nodes transmit the module parameters to the central server. The server then updates parameters for modules shared by multiple nodes (i.e. modules handling the same feature or target) using a federated optimization algorithm such as FedAVG (McMahan et al., 2017b) or FedProx (Li et al., 2020), and distributes the updated parameters back to the nodes for the next round of local training (Figure 1 Panel 4.). This process is repeated iteratively until convergence.

### 2.2.3 HORIZONTAL AND VERTICAL FL **WITH** TRAINING INSTANCE OVERLAP

We briefly outline potential strategies for handling training instance overlap, while leaving a detailed analysis for future work. We assume the same setting as previously, where each local dataset has access to the target values of interest. The simplest solution is to train the model as if no overlap exists, which is reasonable if overlap sizes are negligible. Another approach, suited to a collaborative peer-to-peer setup without a central server, is to train separate local models while securely exchanging intermediate encoded patient states. Each node locally encodes its features and shares the encoded states with nodes holding the same patients; since encoders can be applied in any order, receiving nodes can update and decode the states as needed. This way, each node retains its own model parameters while leveraging shared patient states, which is especially useful when the overlap is substantial. Finally, hybrid approaches that share both patient states and module parameters, or leverage states of similar patients, could be investigated depending on the context. Naturally, ensuring secure data sharing is critical to prevent information leakage in all scenarios (Castiglia et al., 2022; Liu et al., 2024).

## 3 EXPERIMENTS & RESULTS

We conducted three representative experiments to evaluate *FedMoDN*'s robustness against missing data, limited feature subset interoperability, and systematic bias. To simulate realistic clinical challenges, we generated synthetic data, and we used subsets of the California Housing dataset[1] to demonstrate performance on publicly available data. We assessed model performance across various regression tasks. Details on the synthetic data generation process and the public dataset are provided in A.1.1 and A.1.2 respectively. We simulated FL with 10 synthetic nodes and 5 nodes for the public dataset. We emphasized experiments in small data regimes (i.e. around 100 samples per node and 5-10 input features), to reflect real-world scenarios where clinicians train and deploy models on tablets in low-resource settings. In such contexts, mitigating bias and overfitting is particularly important.

**Models** We compared *FedMoDN* to two baselines: (1) *CBaseline*, an upper baseline in which *MoDN* is trained centrally on the entire dataset (i.e. data is shared among nodes), and (2) *FedBaseline*, a federated MLP approach. Like most deep learning architectures, MLPs require complete data, making imputation necessary for missing or incompatible features. We applied mean-value imputation to handle these cases. Alternatively, one could discard incomplete datapoints, but performance would suffer as missingness increases. Moreover, in limited-interoperability scenarios, this could leave some nodes with empty datasets, making the approach infeasible. Details on model architectures are provided in A.1.

**Evaluation** We evaluated each model on 5 different train/validation/test splits derived from each dataset. To ensure a fair comparison, we allowed similar capacities for *FedMoDN* and *FedBaseline*, to achieve comparable performance on fully interoperable, unbiased, and complete data, serving as a reference for subsequent experiments. Training and architecture details are provided in appendix A.1. We measured performance by computing the Root Mean Square Error (RMSE) on the test set and compared the errors of the different models against *FedMoDN* using paired $t-$tests. For selected experiments, we assessed the impact on different types of nodes separately, particularly focusing on how unreliable nodes influence federated training and affect reliable nodes. We observe similar trends on both datasets and present the synthetic data results in the main text, with figures for the public dataset in the Appendix.

---

[1]`https://scikit-learn.org/stable/modules/generated/sklearn.datasets.fetch_california_housing.html`

## 3.1 Impact of Missing Data

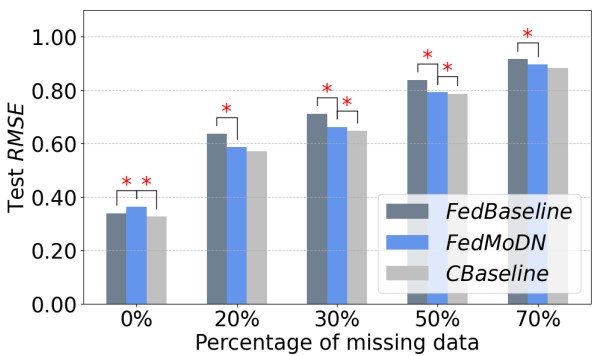

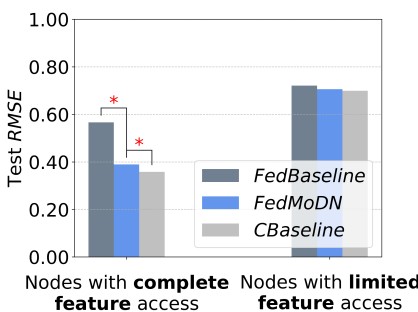

(a) Impact of missing data on model performance. We introduce varying percentages of missing data and compare the results using RMSE. *FedMoDN* consistently outperforms *FedBaseline* as soon as missingness is introduced. While the centralized baseline generally slightly outperforms the federated training, the performance gap remains minimal.

(b) Impact of limited feature interoperability on model performance. We show the RMSE for nodes with full feature access in their train/test sets and those restricted to a limited subset of features. The inclusion of nodes with limited feature interoperability significantly degrades performance when training with the monolithic *FedBaseline*.

Figure 2: Comparison of model performance under different constraints. (a) Impact of missing data on RMSE. (b) Effect of limited feature interoperability on RMSE.

In this first experiment (Figure 2a), we evaluate the robustness of each approach to missing data, maintaining a consistent missingness level across the train and test sets for each node. *Missing data is a common challenge, often arising from incomplete patient records or resource constraints.*

The first set of bars in Figure 2a (corresponding to 0% missingness) shows the performance on complete, unbiased, and fully interoperable data, serving as a reference for subsequent experiments. In this setting, *FedBaseline* slightly outperforms *FedMoDN*, likely due to its simpler training process. However, as missingness is introduced, reflecting real-world conditions, *FedMoDN* consistently outperforms the monolithic baseline at all levels. This highlights *FedMoDN*'s potential to effectively train a shared model even when some nodes experience substantial missingness. Lastly, *CBaseline* usually performs similarly or only slightly outperforms *FedMoDN*, indicating that the architecture is well suited to be trained in a federated way. Similar observations can be made about Figure 4a in Appendix A.2.1 for the public data.

## 3.2 Feature Interoperability

We examine how limited feature interoperability affects model performance. *Limited feature interoperability is especially relevant in clinical resource-limited settings, where inconsistent feature sets arise due to differences in data collection, equipment availability, and cost constraints.* Understanding its impact on FL is crucial for developing models that can handle heterogeneous data sources. To evaluate this, we design an experiment where half of the nodes are restricted to a random subset of the features, while the others have access to all features. This setup is illustrated in Figure 1, Panel 2, where hospitals **A** and **B** use a limited feature subset, while hospital **C** has access to all features. Feature availability remains consistent across each node's train and test sets, allowing us to separately assess its impact on nodes with full and limited feature access in a federated training setup.

As Figure 2b shows, when training under feature heterogeneity, *FedBaseline* significantly harms nodes with full feature access compared to *FedMoDN*. However, all models perform similarly on nodes with restricted feature access. Notably, on the public dataset, *FedMoDN* significantly outperforms the baseline on both node types (Figure 4b).

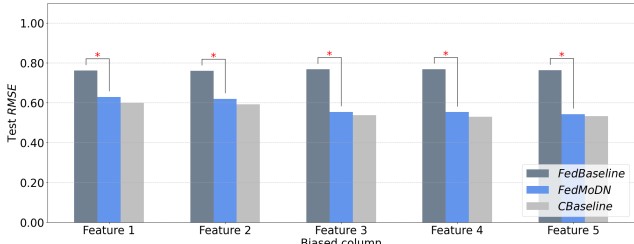

Figure 3: Impact of systematic bias on model performance. Each bar represents the test RMSE for a specific biased feature, using a test set with the corresponding bias pattern. Models are trained with systematic bias in half of the nodes, while a different bias pattern is applied at test time. *FedMoDN* consistently outperforms *FedBaseline*, demonstrating greater resilience to biased feature availability.

## 3.3 ROBUSTNESS TO SYSTEMATIC BIAS

To evaluate each model's robustness to systematic bias, we designed experiments to evaluate how bias in some nodes impacts the overall model. We introduced bias by systematically masking a specific feature in half of the nodes whenever the target variable exceeded the local dataset's average target value. *This simulates a common clinical scenario where certain features are recorded only under specific conditions, creating an unintended correlation between feature availability and patient outcomes.* To test whether the models learn this bias or the true relationships between input and target values, we applied the inverse bias pattern in the test set, masking the value when the target variable is below average. We repeated this process for five different features, creating five distinct biased train and test sets. Figure 3 shows the results across different biased features. *FedMoDN* consistently outperforms *FedBaseline* in each scenario, indicating better resilience to systematic bias. Notably, there is no significant difference between centralized training (*CBaseline*) and federated training (*FedMoDN*). On the public dataset, *FedMoDN* performs similarly or better than *FedBaseline* in every scenario (Figure 5 in Appendix A.2.3). Additionally, we evaluated model performance using FedProx instead of FedAvg, as it is designed to better handle non-IID datasets. However, in this setting, it showed no significant advantage over FedAvg (Figure 6 in Appendix A.2.3). This suggests that the benefits of a modular architecture outweigh those of changing the optimization algorithm in this setting.

## 4 CONCLUSION

In this work, we explored the utility of adapting modular neural networks to complex federated learning settings with both horizontal and vertical partitioning in the presence of bias and missingness. Our experiments highlight the model's robustness to missing data, systematic biases, and limited feature interoperability, challenges that frequently arise when integrating data from heterogeneous sources acquired as a byproduct from existing decision support tools. Moreover, *FedMoDN* offers a promising solution for vertical FL, both with and without training instance overlap, as it does not require discarding heterogeneous data. This is especially beneficial in low-resource settings, where data is already scarce.

### 4.1 FUTURE WORK

As a proof of concept, this work opens opportunities for many extensions and further investigations, both in terms of applications and collaborative learning strategies. First, we plan to evaluate our framework on real-world clinical data, where limited interoperability is a common challenge. In particular, we will explore how feature and target distribution shifts can be optimally handled, for example, through local fine-tuning of federated modules. Then, we aim to explore optimal strategies for training *FedMoDN* in a fully decentralized setting, allowing training instance overlap while eliminating the need for a centralized server. In particular, we plan to investigate how nodes can share patient states to enhance learning without compromising sensitive patient information. Lastly, we plan to extend this work to support multimodal IIO settings where state size optimization would be particularly challenging.

ACKNOWLEDGEMENTS

The authors thank Jonathan Doenz for his feedback on the manuscript.

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

## A APPENDIX

### A.1 MODEL ARCHITECTURE AND TRAINING

*FedMoDN*, *FedBaseline*, and *CBaseline* share simple architectures with 1–2 hidden layers in the encoder/decoder modules and internal layers of the monolithic network, each containing 16–32 neurons. To ensure a fair comparison, we set the state dimensions to 4 and 5 for the synthetic/public datasets respectively, and included a bottleneck layer of the same dimension in the monolithic NN.

The optimal architecture and hyperparameters were selected using cross-validation on the complete, unbiased, and fully interoperable datasets.

We trained the models for $5 - 10$ local epochs for the synthetic/public datasets respectively before each federated aggregation round. Unless specified otherwise, our method follows the standard FedAvg framework. We applied early stopping based on validation data to determine the optimal number of aggregation rounds.

### A.1.1 SYNTHETIC DATA GENERATION PROCESS

For each node, we independently sample data points to generate feature vectors $x \in \mathbb{R}^{10}$ from a multivariate normal distribution with zero mean and identity covariance. The continuous target variable $y$ is then computed as a nonlinear function of $x$, incorporating sinusoidal transformations, quadratic and cubic terms, and feature interactions. To introduce variability, we assign small weights to different components of the transformation. Finally, we add Gaussian noise to simulate real-world uncertainty.

### A.1.2 CALIFORNIA HOUSING DATASET

The California Housing dataset is derived from the 1990 US Census and includes features describing median income, age, and household characteristics such as the number of rooms in Californian houses. The target variable is the median house value. It is publicly available at `https://scikit-learn.org/stable/modules/generated/sklearn.datasets.fetch_california_housing.html`. Given the dataset's large size ($\approx 20,000$ data points), we randomly sampled 100 datapoints per node to fit our experimental setting. The selected features include: ['longitude', 'latitude', 'housing_median_age', 'total_rooms', 'total_bedrooms', 'population', 'households', 'median_income']. We applied standard preprocessing steps, including scaling continuous features, one-hot encoding categorical variables and scaling the target variable.

## A.2 EXPERIMENTS & RESULTS ON CALIFORNIA DATASET

We present here the results of the experiments on the public California dataset.

### A.2.1 IMPACT OF MISSING DATA

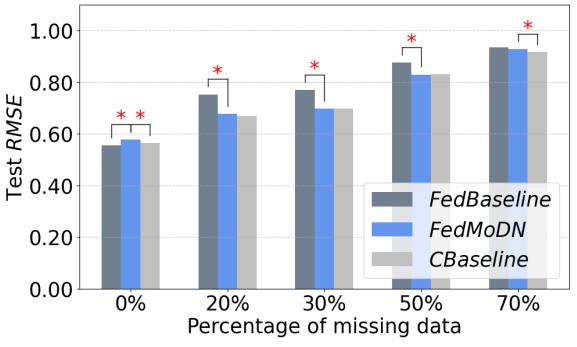

(a) Impact of missingness on model performance. We introduce different percentages of missing data and compare the results using RMSE. *FedMoDN* almost always significantly outperforms *FedBaseline*.

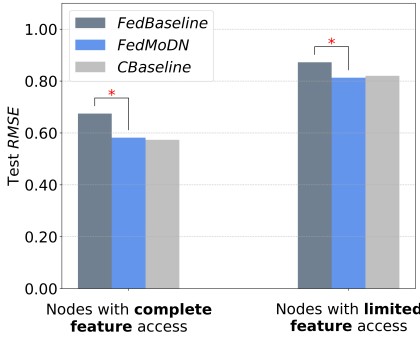

(b) Impact of limited feature interoperability on model performance. We show the RMSE for the nodes that have full feature access in their train/test sets and for those restricted to a limited subset of features.

Figure 4: Comparison of model performance under different constraints using the California Housing dataset. (a) Impact of missing data on RMSE. (b) Effect of limited feature interoperability on RMSE.

Figure 4a presents the robustness of each approach to missing data on the California dataset. As in the synthetic data experiments, *FedBaseline* slightly outperforms *FedMoDN* on the complete,

unbiased, and fully interoperable dataset. However, once missingness is introduced, *FedMoDN* consistently achieves superior performance.

### A.2.2 FEATURE INTEROPERABILITY

As described in subsection 3.2, some nodes in this FL experiment have access to only a random subset of features, while others use the full feature set. For the California dataset, 3 out of 5 nodes were limited to a random subset of half the features. Figure 4b shows that *FedMoDN* significantly outperforms *FedBaseline* on both nodes with full feature access and those with restricted access.

### A.2.3 ROBUSTNESS TO SYSTEMATIC BIAS

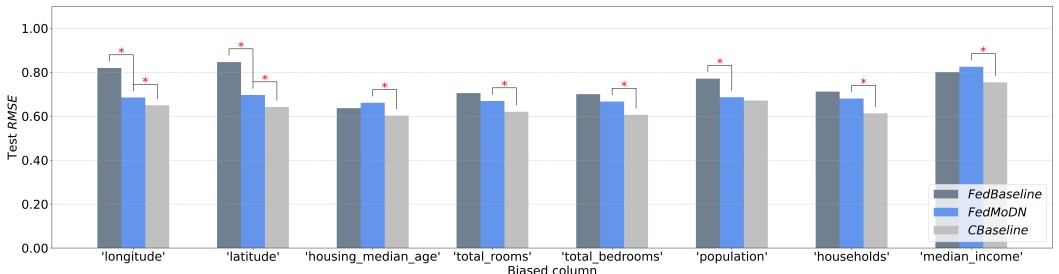

Figure 5: Impact of systematic bias on model performance. Each bar represents the test RMSE for a specific biased feature, using a test set with the corresponding bias pattern. Models are trained with systematic bias in half of the nodes, while a different bias pattern is applied at test time. *FedMoDN* consistently performs similarly or better than *FedBaseline*, demonstrating greater resilience to biased feature availability.

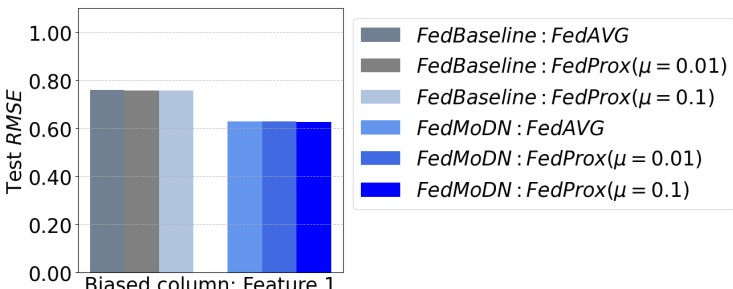

Figure 6: Comparison between FedAvg and FedProx training. Different FedProx settings show no clear advantage over FedAvg on this dataset.

We conducted the same experiments described in subsection 3.3, introducing systematic bias in feature availability and evaluating the models on test sets with different bias patterns. This experiment was repeated for each feature in the California dataset. *FedMoDN* consistently demonstrated greater robustness to bias than *FedBaseline*. Notably, performance varied across features; for instance, *FedBaseline* experienced a significant drop when systematic missingness was introduced in features like "longitude" and "latitude".

Lastly, in Figure 6, we tested whether FedProx better handled bias than FedAvg during training. However, we observed no significant difference for either *FedBaseline* or *FedMoDN*, suggesting that the benefits of a modular architecture outweigh those of changing the optimization algorithm in this setting.

