# OpenReview forum: "FedMoDN: Federated Modular Decision Support Networks"
_ICLR.cc/2025/Workshop/MCDC — MCDC @ ICLR 2025_

### Official Review · Reviewer_xcuT · 2025-02-26

**Rating:** 5
**Confidence:** 4
**Fit:** 5

**Summary:**

The paper introduces FedMoDN, a federated learning approach that leverages Modular Decision Support Networks (MoDN) for collaborative learning across distributed datasets with imperfect interoperability. By extending MoDN (Trottet et al., 2023) to the federated learning (FL) setting, the authors assess its effectiveness through experiments simulating realistic challenges in medical data sharing, particularly in resource-limited settings. Results demonstrate that the proposed modular approach outperforms FL methods on monolithic architectures.

**Reason For Giving A Higher Score:**

Clearly articulating the FL-specific contributions beyond simply applying MoDN in a federated setting, along with including additional FL baselines to contextualize performance gains, would strengthen the paper's impact and make the experimental evaluation more convincing.

**Reason For Giving A Lower Score:**

If the paper merely reimplements MoDN in an FL setting without meaningful adaptation or insights into FL-specific challenges, its novelty may be limited, and the contribution is unclear, as the observed benefits could stem from MoDN itself rather than any federated learning-specific advantage.

**Strengths And Weaknesses:**

*Strengths*
- The paper addresses a relevant challenge in FL, particularly in medical data applications.
- Experimental results indicate performance improvements over monolithic FL architectures.


*Weaknesses*
- The MoDN architecture was originally proposed in (Trottet et al., 2023). The main contribution appears to train MoDN via FL, but it is not clear what modifications, if any, were made to adapt it to the federated setting.
- The paper shows that MoDN outperforms MLP when both are trained in FL, but this advantage also holds in centralized training. It is unclear whether the modular approach specifically improves FL performance beyond what is expected from MoDN itself.
- The naming convention could be improved. If FedBaseline is FedAvg, why not call it simply FedAvg and specify which model used?
- Evaluating more federated learning methods could strengthen the analysis.

**Suggestions:**

- Clearly state the technical novelty beyond applying MoDN to an FL setting.
- Discuss whether the improvements stem from federated training or are simply an inherent advantage of MoDN.

---

### Official Review · Reviewer_p4Lg · 2025-02-27

**Rating:** 6
**Confidence:** 3
**Fit:** 4

**Summary:**

The paper extends the MoDN architecture (Trottet et al., 2023) to enable federated training while addressing vertical data partitioning in IIO datasets. Unlike standard federated learning, which requires imputation or discarding missing data, FedMoDN allows hospitals/nodes to train and use only the relevant modules for their local dataset, making it more efficient and flexible.

**Reason For Giving A Higher Score:**

N/A

**Reason For Giving A Lower Score:**

not enough justification of the method, could be better placed in the context of the broader literature. The paper heavily relies on Trottet et al 2023 and does not provide enough detail as a standalone work.

**Strengths And Weaknesses:**

Strengths

* The paper evaluates FedMoDN under different challenges, including missing data, feature interoperability, and systematic bias, providing a well-rounded assessment.

* The study benchmarks FedMoDN against both centralized and federated models, highlighting its strengths and trade-offs.

* The emphasis on low-resource clinical settings reflects practical constraints, making the approach relevant for real-world use.

Weaknesses

* The paper does not provide a detailed computational overhead analysis of FedMoDN compared to FedBaseline, which is crucial for deployment in resource-constrained environments.

* The modular nature of FedMoDN suggests potential for improved interpretability, but the paper does not explore this aspect in detail.

* Significance is assessed but specific tests of significance are not mentioned or their corresponding values

**Suggestions:**

* The study relies heavily on synthetic data, and while the California Housing dataset is publicly available, further validation on real-world clinical datasets would strengthen the findings.

* Exploring model interpretability to understand how modular decisions are made.

* Investigating scalability and communication efficiency in larger FL deployments.

* More justification of the method and comparison with a broader relevant literature

---

### Official Review · Reviewer_6Byv · 2025-03-02

**Rating:** 6
**Confidence:** 3
**Fit:** 4

**Summary:**

The paper introduces a novel federated modular neural network architecture, FedMoDN, which addresses the challenges of learning from imperfectly interoperable distributed datasets. This is well-motivated and particularly relevant in clinical settings where data heterogeneity and missingness are common.

**Reason For Giving A Higher Score:**

The paper introduces a novel federated modular neural network architecture, FedMoDN, which addresses a critical challenge in federated learning: learning from imperfectly interoperable distributed datasets. This well-motivated and have positive practical impact.

**Reason For Giving A Lower Score:**

Refer to the weaknesses.

**Strengths And Weaknesses:**

Strengths:
+ This work is well-motivated and particularly relevant in clinical settings where data heterogeneity and missingness are common.
+ The experimental results demonstrates that the proposed FedMoDN is significantly more robust to missing data compared to monolithic neural networks.
+ The paper is well-written and easy to read.

Weaknesses:
- While the paper presents promising results on synthetic and public datasets, it lacks validation on real-world clinical data.
- The paper compares FedMoDN with a federated MLP baseline and a centralized baseline. However, it would be beneficial to include comparisons with other state-of-the-art federated learning methods, especially those designed for handling heterogeneous data.
- Investigating the scalability of FedMoDN in terms of computational and communication efficiency would provide a more comprehensive understanding of its applicability in large-scale federated learning scenarios.

**Suggestions:**

Refer to the weaknesses.

---

### Official Review · Reviewer_iKKr · 2025-03-04

**Rating:** 5
**Confidence:** 3
**Fit:** 4

**Summary:**

This work proposes FedMoDN, a federated modular neural network designed to handle imperfectly interoperable distributed data (IIO), particularly relevant to medical applications. Unlike traditional federated learning (FL), which often assumes complete feature availability across nodes, FedMoDN allows collaborative learning across vertically partitioned datasets without discarding or imputing missing data. The architecture leverages modular encoders and decoders, enabling institutions with different feature sets to contribute to a shared model without exposing their raw data resulting in a robust model and better performance validated over 2 datasets.

**Reason For Giving A Higher Score:**

The idea is novel, well-motivated and the paper presents valid and impressive results. The analysis follows a justifiable trend and is logically sound.

**Reason For Giving A Lower Score:**

The presentation requires a lot of work, with key experiments central to the work missing.

**Strengths And Weaknesses:**

Strengths:
1. Sufficient experimentation along with adequate explanation are presented with well-documented, meaningful results.
2. Superior handling of systematic bias in data collection is demonstrated.
3 .Robustness when dealing with limited feature interoperability is obtained.

Weaknesses:
1. Medical data and applications are mentioned at various places but no medical datasets are used.
2. While the paper mentions handling scenarios with training instance overlap (where different nodes have data from the same patients), the actual experiments only focus on non-overlapping cases. The proposed approach for overlapping instances is just outlined theoretically without empirical validation
3. The paper compares mainly against a basic MLP with mean imputation rather than against more sophisticated approaches for handling missing data or vertical federated learning
4. Analysis of how sensitive the model is to hyperparameter choices like state vector dimension, which could significantly impact performance is missing.
5. The paper presentation needs work, especially the sectioning of data.

**Suggestions:**

For analysis based suggestions, refer to Weaknesses.

Presentation also needs more work, some key points are:
1. Personal pronouns like "we" should be avoided.
2. Medical data is mentioned but not used anywhere.
3. Mathematical formulation/modelling behind the algorithm is missing.
4. Consider merging or removing some sections in the Method section.

---

### Decision · Program_Chairs · 2025-03-06

**Decision:**

Accept

**Comment:**

The paper critiques current decentralized learning evaluations that assume shared metadata by highlighting a discrepancy between research settings and real-world constraints. The paper recieved borderline score from most of the reviewer and however most of them agree that the paper is good fit for the workshop. We suggest the authors to perform some experiments on real world data along with some additional baseline as suggested by multiple reviewers of authors to strengthen the paper. Despite the comments a workshop paper it is okay for a workshop paper to just present experiments related to the main claim, however, we agree the experiments could be strengthened further. Overall, this is a borderline case and we're recommend to accept this work to the workshop.